# Making Neglect Invisible: A Qualitative Study among Nursing Home Staff in Norway

**DOI:** 10.3390/healthcare11101415

**Published:** 2023-05-12

**Authors:** Stine Borgen Lund, John-Arne Skolbekken, Laura Mosqueda, Wenche Malmedal

**Affiliations:** 1Department of Public Health and Nursing, Norwegian University of Science and Technology (NTNU), 7491 Trondheim, Norway; 2Keck School of Medicine, University of Southern California, Los Angeles, CA 91803, USA

**Keywords:** neglect, nursing home, residential care, long-term care, quality of care, patient safety, nursing staff’s perceptions, qualitative, constructivist grounded theory

## Abstract

Background: Research shows that nursing home residents’ basic care needs are often neglected, potentially resulting in incidents that threaten patients’ safety and quality of care. Nursing staff are at the frontline for identifying such care practices but may also be at the root of the problem. The aim of this study was to generate new knowledge on reporting instances of neglect in nursing homes based on the research question “How is neglect reported and communicated by nursing home staff?” Methods: A qualitative design guided by the principles of constructivist grounded theory was used. The study was based on five focus-group discussions (20 participants) and 10 individual interviews with nursing staff from 17 nursing homes in Norway. Results: Neglect in nursing homes is sometimes invisible due to a combination of personal and organizational factors. Staff may minimize “missed care” and not consider it neglect, so it is not reported. In addition, they may be reluctant to acknowledge or reveal their own or colleagues’ neglectful practices. Conclusion: Neglect of residents in nursing homes may continue to occur if nursing staff’s reporting practices are making neglect invisible, thus proceeding to compromise a resident’s safety and quality of care for the foreseeable future.

## 1. Introduction

A major responsibility of nursing home staff is to ensure that residents receive the care they need safely [1]. Increasing evidence shows that residents in nursing homes do not always receive good nursing care [2,3,4,5,6]. Nursing home staff are at the frontline of identifying care practices that can possibly harm residents, but they may also be at the root of this problem [7,8,9,10]. Such harmful practices or neglect of care needs have been approached from different perspectives [2,3,9,10,11,12,13,14,15,16]. Still, the outcomes of these practices are the same for the residents, regardless of the chosen perspective or labelling, or if they are intentional or not. These neglective practices may cause harm or potential harm to residents, threatening the residents’ safety and overall quality of care.

Studies that have explored the reporting of harmful practices in nursing homes using quantitative measures [2,4,10,17,18]; thus, they did not capture the impact the professional-practice culture in nursing homes has on the reporting of neglective practices. There is a need for studies that investigate reporting practices and explore the circumstances that influence the reporting of neglective practices among nursing home staff. In this paper, we used neglect and neglective practices to describe the spectrum of possible harmful practices occurring in nursing homes. The Centers for Disease Control and Prevention (CDC) defines neglect as “*failure by a caregiver or other person in a trust relationship to protect an elder from harm or the failure to meet needs for essential medical care, nutrition, hydration, hygiene, clothing, basic activities of daily living or shelter, which results in a serious risk of compromised health and/or safety, relative to age, health status, and cultural norms*” p. 34 [19]. Based on this definition, we understand neglect to include any aspect of required resident care that is omitted (either in part or as a whole) or delayed. Hence, basic care of physical needs, emotional/psychological support, and attention to the social needs of each resident [15,20,21] are necessary for good basic care. Neglect includes acts of omission, such as missed nursing care [18], and acts of commission, such as medication errors [22]. According to this understanding, neglective practices are related to both the failure to protect residents from harm and the failure to provide basic care, thus compromising patient safety and quality of care. 

Quality and patient safety are not synonymous but patient safety may be viewed as an essential part of quality care [23]. While patient safety mostly focuses on protecting patients from harm, quality of care may also include other dimensions of care, including quality of life. The latter has personal and professional values, morals, and ethics incorporated [24]. Hence, quality in healthcare is a multidimensional concept, and the delivery of high-quality healthcare is complex, requiring collaboration and a shared understanding of what is included in quality among policymakers, managers, care staff, patients and relatives [25]. This may also vary, depending on the perspective and local context [9,24].

A constructivist grounded theory approach guided our work [26]. We sought to understand the processes that influence the culture of reporting neglect in nursing homes as outcomes of social interactions and processes among the nursing-staff members and managers. We explored processes that shape staff members’ perceptions and behaviours regarding this reporting. The aim of this paper was to generate new knowledge on the reporting of neglect in nursing homes, based on the research question “How is neglect reported and communicated by nursing home staff?” 

### 1.1. Reporting of Incidents in Healthcare

To prevent harmful practices in healthcare, there is a need for comprehensive information about the care provided [27]. Quality and risk management systems are essential to systematically identify and assess potential threats to patient safety and inadequate quality of care [28]. These systems depend on reports submitted by frontline staff that contain data on potentially harmful practices that threaten patient safety and quality of care. A main goal of these systems is to identify risky healthcare practices. Registering and monitoring harmful practices are deemed to be pivotal for organizational learning and preventive measures and as essential in improving the quality of care [25,29,30,31]. A well-functioning incident reporting system (IRS) must be confidential, blame free, easily accessible, and contain constructive feedback [28,32]. The WHO defines an “incident” as “*any deviation from the usual care that poses a risk of harm or causes injury to a patient; it includes errors, preventable adverse events, and hazards*” p. xii [32]. Hence, in this paper, we use the term incident, and we also use the term deviation, related to deviation from usual care, which is a common term used in Norwegian healthcare and also by our participants. 

### 1.2. Reporting Culture in Nursing Homes

The complexity of nursing home residents’ care needs, combined with heavy workloads and a lack of resources, increase the risk of neglectful practices [33]. The importance of establishing a system and culture that encourages the reporting of incidents in nursing homes has been emphasized [34]. Examples of incidents reported in an IRS system in nursing homes vary from defective technical equipment, medication errors, procedural failures, falls, aggression from residents, failure to observe deterioration in critical illness, development of pressure ulcers, and missed basic hygiene. Most incidents reported by staff in nursing homes are medication or procedure related [8,11,35,36]. Furthermore, there is an underreporting of incidents in nursing homes [27], including the neglect of residents’ care needs [8,9,35,37]. Underreporting is partially due to both systemic factors, such as lack of time, complicated reporting systems, missing guidelines for reporting, and a lack of feedback and reaction [8,25,38]. At the individual and cultural level, a lack of reflection on own practice, loyalty among staff, fear of conflict with colleagues, and a blaming culture have been found to be important factors [7,8,9,27,35,36,38,39,40,41,42]. In a recent review, it was found that fear of negative consequences is the main barrier to incident reporting by nurses [25]. 

### 1.3. Reporting Practice in Norwegian Nursing Homes

Neglect in nursing homes may lead to severe injury or death. According to Norwegian law, “§196 of the Penal Code”, all citizens have a duty to prevent a criminal offence by reporting it to the police. This duty to protect nursing home residents supersedes the duty of confidentiality [43]. There is currently an ongoing debate about the establishment of mandatory reporting of elder abuse and neglect in nursing homes. Healthcare providers’ responsibility to report threats to patient safety is formally regulated in the National Health Personnel Act § 17; “*Health personnel shall of their own accord provide information to the supervising authorities on matters that may endanger patient safety*” [44]. In addition to central regulations, healthcare personnel have a legal and ethical responsibility to protect patients’ safety. The ICN code of ethics for nurses states that “*Nurses facilitate a culture of safety in health care environments, recognising and addressing threats to people and safe care in health practices, services and settings.*” p. 8 [45]. 

With the exception of incidents that threaten residents’ safety or the quality of services provided, there is no formal obligation to report incidents to national health authorities [46]. Norwegian nursing homes are thus mandated to have an internal quality control system to monitor patient safety and overall quality of care [47]. In accordance with this mandate, nursing homes register certain measurable indicators related to the structural characteristics of quality of care. These are related to reviews of medication, the prevalence of infections, injuries from falls, and screening for risk of malnutrition [24]. Incidents related to these quality indicators and other deviations from usual care are reported in internal control systems such as IRS, also known as a deviation system. The Working Environment Act protects employees from eventual negative sanctions and reprisals related to the reporting of incidents. This includes any unfavourable act, practice, or omission that is a consequence of or a reaction to the fact that the employee has given notification of incidents [48]. The aim of this paper was to generate new knowledge on the reporting of neglect in nursing homes, based on the research question “How is neglect reported and communicated by nursing home staff?”

## 2. Materials and Methods 

### 2.1. Research Design

Based on the aim and research question, a qualitative design was chosen. Qualitative methodology is used in the exploration of meanings of social worlds experienced by individuals in their natural context. It can give insights into people`s social experiences, thoughts, expectations, meanings, attitudes, and communications related to interaction, relation, development, and interpretation. A constructivist grounded theory (CGT) approach was taken, which emanates from the idea that interactions between people create new insights and knowledge. CGT is a flexible but also structured method that involves simultaneous data-collection and analysis. It uses the comparative method and can provide tools for constructing theory. In keeping with CGT, data were collected and analysed in an iterative process [26]. Focus groups (FG) were initially chosen for logistical reasons, but also for their potential to produce rich data. Focus groups are used to find a range of reflections from people across several groups, and this method can provide insight into specific themes of interest. The group processes can produce a synergy that individual interviews alone do not produce. Group dynamics can create new perspectives and knowledge, which can be further developed through the CGT approach [49]. Due to the sensitive topic and the possibility of a negative effect of group interactions, we supplemented the focus groups with individual interviews (II). 

### 2.2. Participants

A total of 30 participants (age 22–62 years, work experience in nursing homes 1–28 years, 27 females/3 males) were recruited from 17 nursing homes (four rural and 13 urban) in Central Norway, all with experience in caring for long-term nursing home residents. Participants were provided with an invitation letter about the research project and research team as a part of the recruiting process. Participants were recruited in different ways; through nursing home managers, educational settings, and individual gatekeepers, during a 17-month period from April 2019 to November 2020. Delays and complications in recruitment due to lockdown during the early stages of the COVID-19 pandemic were experienced from March 2020 onwards. Five focus-group discussions and 10 individual interviews were conducted. A more detailed overview of the participants and interview data is available in a previous publication from this project [16].

### 2.3. Data Collection

The interviews lasted 1–1.5 h and were guided by a semi-structured interview guide, which was developed and adjusted in line with our initial analyses (Additional Appendix A). After the participants’ spontaneous responses had been explored, we introduced case descriptions and statements about elder neglect from a survey instrument on elder abuse in Norwegian nursing homes [2]. During the initial analyses, we moved on from focus-group discussions to individual interviews to create a setting that could provide further insight into potentially sensitive issues. This strategy did not initially add significantly to the already frequently reappearing themes explored in the focus-group discussions. However, additional categories were developed after case descriptions and examples of neglect from a survey instrument on elder abuse were introduced [2]. In accordance with the CGT approach and for an even deeper insight into the developing concepts, we strategically selected participants who had chosen to no longer work in a nursing home. They were reached without institutional gatekeepers, thus creating more variation in the sample and enabling us to generate more conceptual categories. Interviews were digitally recorded and transcribed verbatim by an experienced transcriber (HF). This study was conducted according to the guidelines of the Declaration of Helsinki and approved by The Norwegian Centre for Research Data (NSD) (protocol code 221320, approved 26 February 2019). We used the COREQ checklist to ensure methodological quality (Additional Appendix A) [50].

### 2.4. Data Analysis

The transcripts were initially analysed line by line by the first author with the use of pen and paper, which were supplemented by independent contributions from the second and last author, using initial, focused, and theoretical coding, as guided by the CGT. Concepts and categories were developed through constant comparisons to identify variations and consistencies in the material. Field reports and memos were produced throughout various phases of data collection and analysis, ensuring an audit trail. NVivo software version 20 was used to organise the data. An example of the coding process is presented in Table 1.

Several steps were taken during the process of data collection and analysis to develop and explore the analytic categories. However, after five focus-group discussions and 10 individual interviews, gathering more data did not seem to introduce new properties or provide further insights about our categories. Although we missed some of the criteria for a more formalised theory, we found our data and analysis to be sufficient.

## 3. Results 

Our core finding is the existence of a reporting culture among nursing home staff that renders neglectful practices invisible. We present some of the nursing home staff practices that make neglect invisible below in Figure 1. These practices occur when nursing staff (1) omit to report neglective practices, and when they (2) practice staff protection.

### 3.1. Omit to Report Neglective Practices

The category ‘omit to report neglective practices’ has two subcategories, namely deprioritizing written reporting and perceiving neglect as missed care.

#### 3.1.1. Deprioritizing Written Reporting

Certain omissions indisputably belong to the list of (in)actions that should be reported in the IRS, such as those related to resident procedures, medication administration, or (in)action resulting in resident injury or potential injury. The participants strongly agreed that incidents that pose a potential “*threat to the life and health of the resident*” should be reported promptly. There were, however, situations related to care that were not as clear as these episodes and may be open to personal and subjective judgement by nursing staff, as shown in the following statements: 


*“In other words, it is a deviation when things are not done.”*

*(FG, RN 1)*



*“If there is something we react to, we report this as deviations.”*

*(FG, RN 7)*



*“It is so difficult to define. When is it deviation and when is it neglect?”*

*(II, A 3)*


As illustrated by these statements, these situations were open to different interpretations, depending on the eye of the beholder. There may be different thresholds for which (in)actions triggers a reaction, or how the individual carer interprets tasks that are not properly done. Hence, there was interpersonal variability in what was perceived as a deviation that qualifies to be reported in the IRS. This was not the only reason for not reporting neglectful practices in the IRS. Another relevant factor was the ambiguity between deviation and neglect. This ambiguity may reflect a lack of knowledge of what qualifies as neglect, but also differences of opinion as to what counts as neglect. Consequently, deviations not recognized as neglect were hence not reported in the IRS. The above statement illustrates that there was a perceived difference between deviation and neglect and that they were handled differently. 

There were deviations that were judged as not serious enough to be reported in the IRS. These were typically tasks that are not clearly visible if they are omitted, such as not providing personal hygiene or mouthcare regularly. Ideally, they should equate incidents such as the development of a pressure ulcer and not providing mouthcare, but, in practice, these two instances were treated differently in relation to observation, reporting, and preventive measures. It could also be episodes of omissions that were perceived to have no or only minor consequences for the resident. If an incident did not lead to any major consequences for the resident, it tended to be only communicated verbally, even if it was related to the administration of medication.


*“I can say to the person I know has administered the medication that, you know, I found this and this [errors]. I haven’t written any deviation on it, but you should be more careful next time.”*

*(FG, RN 7)*


This resembled their discussions of the omission of other care activities: 


*“-if we forget to put on some glasses or hearing aids, we know that it may not have any consequences.”*

*(II, LPN 1)*


The main reason given for not reporting in the IRS was lack of time, which was mainly related to a heavy workload with an overload of care activities, leaving little time for incident reporting. In addition, the IRS was complicated and time consuming to navigate. 


*“The deviations usually occur due to lack of time, and then you must take the time to sit down and write the deviation report. And you are not allowed to write overtime work because you do not get paid for it anyway. So, you can’t bear to sit at work for free and write deviation reports [laughs].”*

*(FG, RN 1)*


Consequently, verbal feedback on reporting of neglective practices was preferred over written reporting. Hence, episodes of neglect were communicated to other staff or management, but preferably verbally and informally, resulting in zero documentation in the IRS. The practice of verbal communication of neglect was questioned by one participant because *“this makes the omission disappear”*. These factors lead nursing staff to deprioritize written incident reporting, although they acknowledged such reporting as an important tool for improved safety and quality of care. 

#### 3.1.2. Perceiving Neglect as Missed Care

As mentioned previously, there was a broad agreement that neglectful practices posing a possible threat to a resident’s life and health should be documented in the IRS. Other, more subtle omissions, including activities related to personal care such as not changing pads, not putting lotion on fragile skin, or not meeting a resident’s psychosocial needs, were subject to more individual considerations and were unlikely to be reported. These missed care activities were frequently noted in patient records or other informal systems such as a “to-do list”, and, therefore, they were not necessary to report in the IRS. These omissions were not visible in the same way as medication or procedural errors, falls, or pressure ulcers, which were commonly documented in the IRS. 


*“There are many things that you should write a report on that we don’t; we are mostly focused on writing a deviation report when there is a fall or medication errors.”*

*(FG, LPN 1)*



*“No, I don’t think so. I believe that they think that it is actually a deviation, that one should perhaps have done it [written a report]. I do not know [referring to not changing pads at night].”*

*(II, LPN 9)*


Hence, these episodes of neglect were not reported in the IRS because they may be labelled as missed care, meaning that these omissions could *“be done later”.* This nonreporting was a regular occurrence when postponed care activities were eventually missed altogether and perceived as minor incidents.


*“All those things [regarding personal hygiene] can appear on the daily record keeping, but not on any status report or any deviation record.”*

*(FG, RN 11)*



*“RN 2: No, then I think that [the shower] I’ll do it another day. RN 1: Yes, we can do that tomorrow. It’s no big deal. RN 2: No. RN 3: Or we sort of pass it on to the to-do list for the next day. But then, if it is not carried out this week [laughs], we will not write a deviation on it.”*

*(FG, RN 1, 2, 3)*


This practice of not reporting missed care activities was also supported by managers who were hesitant to have frequent episodes related to omitted personal care reported in the IRS, but rather preferred to focus on reporting more *“important incidents”*.


*“...if someone repeatedly didn’t get a shower or things like that for a period, we wrote deviation reports on it, but then we kind of got a reprimand from the bosses. Their argument [the management] was that it was not serious enough and that you can just leave it [the task] until the next day. But then again, you can say it, but it will not be done the next day, because then it is someone else’s turn to shower.”*

*(II, RN 9)*


The frequency of the neglectful situations matters, as there is a difference in whether care is missed for one day or for a week. A minor omission perceived as missed care, and, thus, only reported verbally, may become more serious when it occurs repeatedly or systematically. These dilemmas were acknowledged by participants who expressed that they miss more discussions and reflections among colleagues about the care they provide and where, how, and when to acknowledge missed care as neglectful practices. Due to these discrepancies, many episodes of neglect did go unreported in the IRS.

### 3.2. Practicing Staff Protection 

This category had two subcategories, reluctance to indicate or reveal neglectful practices and using nonconfrontational strategies.

#### 3.2.1. Reluctance to Indicate or Reveal Neglective Practices

There are different ways of providing care in a nursing home, and nursing staff described an open culture where different care practices are discussed one on one among staff or in plenary sessions. What constitutes adequate care may thus be reflected upon in group meetings. Nevertheless, there were situations where individual staff members considered their own or a colleague’s practice as inadequate. These episodes were not necessarily communicated or reported. A major reason for this reluctance was that they want to avoid confrontations with colleagues. This was related to a work culture where they experience or fear unpredictable or negative responses from colleagues, such as aggression or sadness when reporting neglectful practices.


*“The best would be to go to that person and talk to him/her, but it’s a bit unpopular to do that. I speak for myself, I’m not tough enough to do it because it’s uncomfortable.*

*(FG, LPN 5)*



*“...but there are some who are like “are you saying I’m not doing my job?”*

*(II, LPN 7)*


Although there was an agreement that reporting incidents of neglect is a way to raise awareness and possibly improve poor practices, this is not easy to execute. Reporting neglectful practices made nursing staff feel that they are turning their colleagues in, leaving an extremely unpleasant feeling. It made nursing staff feel disloyal as if they were betraying their colleagues when revealing neglectful practices to managers and other colleagues.


*“No, you’re kind of together “by the bedside”. A bit like that, should I snitch on someone?”*

*(II, LPN 10)*


They also described that incident reporting can exacerbate a harassment culture, where staff take turns at reporting to harass each other. In addition, they were reluctant to report neglectful practices because they knew that they would do similar things themselves.


*“.. but they’ve said that too, because if you write a report on me, you are after me, aren’t you?”*

*(II, LPN 7)*


Hence, nursing staff found it uncomfortable and unpleasant to reveal their own neglective practices related to the quality of care they are able to provide. In a work setting where there is a continuous lack of time and an overload of work activities, nursing staff knew that they will not be able to fulfil all their duties or do all the tasks they are supposed to.

#### 3.2.2. Using Non-Confrontational Strategies 

Talking directly to colleagues when observing neglective practices was not a problem in nursing homes with an open reporting culture. Providing direct feedback was easier when staff knew each other well and could predict how the message would be received, and when they have a culture of communicating with respect and curiosity. Feedback based on professional knowledge may also be easier to give (and receive), but personal self-esteem and life wisdom also gave nursing staff the confidence to report directly to a colleague. 

However, the opposite was also noted. It was difficult and problematic to communicate, and report observed neglect by a colleague in a direct and clear way. The main reason for the difficulties in communicating and reporting neglective practices was an intense feeling of discomfort experienced in these situations. Consequently, the staff steered clear of this discomfort by applying different strategies to avoid direct confrontation with a colleague over neglectful practices. 


*“It is perhaps easier to tell the manager, because then they can notify the person of concern. Because it becomes a bit like, what right or duty do I have as a regular employee to criticize someone at the same level as myself?”*

*(FG, RN 1)*


In this process, hierarchical structures, ranked according to experience, education level, age, and responsibility, mattered. Nevertheless, discrepancies in the staff’s perceptions were observed; some found it more difficult to confront a colleague at a higher level in the hierarchy, whilst others found it more disturbing to confront a colleague at the same level as themselves. To avoid direct confrontation with a colleague at the same level in the hierarchy, reporting to someone higher up, for example, group leaders or management, in writing or verbally, was preferred. This also included written incident reports, where the responsibility for reacting to and following-up on the omissions were transferred to the manager. By reporting upward in the organization, they had done their “duty” without having to deal directly with possible negative responses. 

Another way to avoid direct confrontation was by means of general feedback in plenary meetings organized by a group leader or management. This could also be addressed in a general manner, avoiding identifying a specific individual. As there was no guarantee that the message would reach the intended recipient, feedback in plenary sessions were regarded as having some drawbacks. 


*“That’s probably what is done most, and it’s also much easier [laughs] than criticizing someone—*
*or not criticizing—*
*but giving feedback like that to just one person. It is easier to rather say it in public [on more general basis]. Because then you don’t feel that you are attacking the other person.”*

*(FG, RN 1)*



*“I don’t have an overview of all the deviations, but it is brought up at the staff meeting how many deviations there have been and that we need to focus on how important it is to check medication prescriptions when giving medication.”*

*(II, RN 13)*


The nursing staff had different creative solutions to communicate neglectful practices without directly criticizing their colleagues’ practice, for example, “sugar-coating” feedback or camouflaging feedback as advice or an offer to help.


*“Yes… I might sometimes pretend to give advice [laughs]. That “it might be a good idea to...” [laughter] … not as criticism, but as advice.”*

*(FG, RN 1)*


Intentional neglect was uncommon and easily excused through attributions of lack of time, competencies/training, or experience. Wanting to believe that their colleagues are trying to do their best and that observed neglectful practices are unintentional, they chose to naïvely overlook neglectful practices.


*“I choose to believe that you don’t see it, that you don’t think about it and that you don’t see the whole as you should. I can’t believe that someone wilfully… think I can’t bear that if…, I can’t bring myself to believe that.”*

*(FG, RN 10)*



*“There are colleagues you work together with many days a week. So, you’re a bit afraid of creating a bad atmosphere… [laughs], yes. Also, you probably have such good faith in people that you believe that they don’t do this intentionally. So, it’s hard to bring it up.”*

*(FG, RN 1)*


When staff *“chose to believe the best”* and did not recognize or acknowledge colleagues’ practices as neglective, this would not be reported, thereby avoiding a difficult confrontation or situation for themselves and their colleagues. Still, questions were raised as to whether neglect might happen more frequently than they wanted to believe. There were instances of awakening to this possibility during the focus groups/interviews, implying that there can be different degrees of this and that it should be reported.

The interviews triggered reflections among the staff and broadened their perspectives regarding their own practices. Hence, it was acknowledged that the nursing staff should be more aware of the IRS, as there is room for improvement regarding the reporting of neglectful practices.

## 4. Discussion

Overall, our participants were positive about reporting neglectful practices. Their main reason for reporting neglect was to raise awareness and improve practice, with the acknowledgement that neglectful practices could possibly result in incidents and, therefore, there should be a prompt response when neglect is discovered. Nevertheless, we found situations where the nursing staff did not do this in practice. Detecting, reporting, and, possibly, preventing neglective practices that can potentially lead to incidents are crucial to improve the safety and quality of care for residents in nursing homes. Hence, knowledge about circumstances that influence the nursing staff’s decisions and preferences regarding whether to report neglect is highly sought after and required. In the following, we discuss some structural, individual, and sociocultural issues that may influence and possibly prevent nursing staff from reporting neglective practices.

### 4.1. Neglecting Existing Reporting System

Our findings are similar to those of previous studies where heavy workloads and lack of time to report in the IRS are major barriers to reporting neglectful practices [8]. In addition, we found other structural challenges to the reporting of neglect in nursing homes. In Norway, there is no mandatory reporting system for incidents in nursing homes, and each institution must address this issue as a part of the internal quality-control system. The quality indicators in the current quality systems are mainly focused on objective, measurable indicators of care, such as medication errors, the prevalence of infections, and malnutrition [51]. Prior research found that these quality indicators do not cover the omissions related to basic care, and especially the subtle incidences, such as neglecting psychological or social care [39]. Without a suitable system for the reporting of neglective practices, it will be challenging to create a culture of reporting neglect in nursing homes [24,29]. 

In accordance with previous research, we found a preference for face-to-face interactions [8] and verbal reporting [52] over written documentation. This is also supported by Moore, who questions these practices of informal reporting and asks whether written, more formal reporting, should be encouraged [42]. However, informal and interpersonal communication about concerns regarding suboptimal care has been found to be a subtle but effective way of confronting colleagues’ behaviours and is equally effective as whistleblowing or other formal reporting strategies [52]. These informal practices have an immediate and powerful effect not seen in the formal, more bureaucratic, and distanced reporting systems. Nevertheless, these informal strategies for raising concern remain unrecorded and invisible for monitoring, both internally and externally [52]. 

Thus, an IRS system is perceived as a crucial tool to improve practice, even though the reporting practices found in this study challenge the role of the IRS in this matter. To summarize, when nursing staff do not use an existing reporting system and prefer informal and verbal over formal and written reporting, neglect may continue to be hidden or invisible, and a challenge for resident safety and quality of care in nursing homes. 

### 4.2. Acknowledging Neglect—A Prerequisite for Reporting

Our participants questioned what constitutes neglect versus deviations and also which episodes that were acknowledged as important enough to be reported in the IRS. There was strong agreement among our participants that incidents that pose a “*threat to the life and health of the resident*” should be reported promptly. Situations that are perceived as threatening patient safety were reported in the IRS without further ado. This was typically the episodes that are visible and difficult to overlook, such as medication errors, fall injuries, or other instances of physical harm to a resident. More subtle incidents and incidents that do not cause immediate harm were easier to overlook, also due to the fact that there is less-visible evidence of neglect. We found that potentially severe incidents that caused little or no harm were often not reported in the IRS, especially if these were related to individual care failures. 

Our findings align with previous research that has demonstrated that a major reason for the underreporting of neglect in nursing homes is a poor understanding of the terminology and a lack of a standardized definition. The interpretation of the terminology varies across the cultures, countries, settings, and perspectives of those involved [1,8,10,11,14,53]. In addition, there are different views concerning the degree of severity of the consequences [8,9,10]. Hence, a lack of knowledge and ambiguity as to what constitutes neglect have implications for both the detection and reporting of neglect [7,8,10,14,16,35,54,55].

In addition, certain norms and attitudes among nursing home staff and managers, such as the normalization or legitimizing of neglect [14,16,56], neglect as a part of nursing home culture [5,57,58] or necessity prioritization [39] also contribute to not recognizing or reporting neglect. The tendency to accept and legitimize neglect also develops when nursing home staff perceive neglect as missed care [16]. In these cases reporting of neglect was omitted because of an acceptance among staff that missed care activities can be postponed—although, in many cases, this meant that they would not be done anyway because there is not enough time later [16]. Hence, nursing home staff tended to not acknowledge their missed care practices as neglect or make themselves accountable for these omissions by reporting it in the IRS. The nursing staff’s ambiguous perception of neglect and deviations from the usual basic care makes it more likely that these neglectful practices are not reported in the IRS. Thus, neglect becomes an acceptable practice in nursing homes, and awareness of their own practice and the possibility of improving the quality of care is at stake. 

### 4.3. Reinforcing the “Blame and Shame” Culture

Our findings are in line with prior research that has demonstrated that one of the barriers to the reporting of neglect in nursing homes is loyalty among colleagues [8,33,35,42,55]. We found that nursing staff prefer to collaborate and deal with deviations from the usual care at the same level in the hierarchy. Not reporting observed neglect by colleagues became a way of “having each other’s backs”, as staff realize that they are likely to also neglect residents’ needs due to the well-known challenges of tasks and time. Nursing staff being unable to provide basic care is challenging the fundamental notion of nursing. The realization that their own care practices, or lack thereof, might be harming the residents, threaten their professional self-esteem, and may cause moral distress [18,40,45,58]. Thus, when nursing staff have to compromise, they are faced with failing their own ideals for care provision [40]. This may create a culture of individual shame for carers who are unable to meet their own personal and professional standards [59]. In addition, fear of more formal and informal sanctions may come into play through the omission to report their own neglective practices, confirming their own shortcomings in doing their work [40]. 

The possibility of a more formal reaction and negative consequences is a well-known challenge for whistle-blowers [25,31,32,40]. A blaming culture may exacerbate a toxic work environment, making it even more difficult to report neglect [40]. Norway has a relatively open and non-hierarchical healthcare system, with laws that formally protect whistle-blowers compared to some other countries where one may be held personally accountable and risk facing a lawsuit when reporting individual or system failures [60]. Nevertheless, persistent challenges with underreporting in Norwegian nursing homes may lead to the possible introduction of anonymous reporting, which has been implemented successfully in some countries [9,54]. This strategy, which intends to protect the staff, can be questionable, and it may be argued that patient safety should be the main priority. Others have stressed the importance of a legal framework for reporting practices, based on a system approach without undermining personal accountability [29]. 

A “blame and shame” culture enables nursing staff to conceal neglectful practices, whether performed by themselves or colleagues. A workplace culture that suppresses openness can, over time, normalise the deviation from usual care. Hence, there is a relationship between worker safety and patient safety that may also influence the quality of care [52]. Concealing errors and incidents may lead to inaccurate interpretations and evaluations and decreases the possibility of learning from errors, making the prevention of incidents more difficult [25,38]. 

### 4.4. Refusing to Confront Colleagues

The strong aversion to confronting colleagues may be related to the nursing staff’s need for conformity, which makes it difficult to report suboptimal care performed by colleagues [58]. In addition, a working environment that makes them dependent on their colleagues can be a reason for refusing confrontation [40,41]. Several researchers have found that staff admit that they are not brave enough and need more psychological safety to dare to confront colleagues about neglectful care [9,60]. A lack of or a negative response from the management or colleagues further undermines the desire to provide formal or informal feedback of neglective practices in their workplace [31,40,41,61]. Jones and Kelly found that reporting is regarded as a high-risk activity with few benefits for nursing staff [52]. Thus, snitching or whistleblowing on colleagues is regarded as a last resort, and nursing staff prefer subtler and less-confrontational ways of reporting substandard care [8,52].

Hence, we found a diverse repertoire of non-confrontational strategies that are used to communicate neglect among nursing staff. A common non-confrontational strategy used by nursing home staff is “sugar-coating” or camouflaging feedback on colleagues’ neglective practices as advice or offer to help. Similar strategies have been described by Jones and Kelly, who found that nursing staff used interpersonal strategies, such as humour or sarcasm, to show their dissatisfaction with colleagues’ care [52]. Prior research has found disengagement [40] or cognitive dissonance [58] as strategies that prevent neglect being reported. As shown in our analysis and confirmed by others, this occurs when nursing staff choose to overlook neglect and believe the best of their colleagues [39] or they choose to notify management in a more informal way, omitting any written recording of what they had discerned. These non-confrontational strategies make it unnecessary to report, thus indicating that their professional and personal standards for care are just “stirred, not shaken” enough to make them report neglect. 

Practising non-confrontational strategies may further increase tolerance for neglectful care practices. Hence, unacceptable behaviour becomes normalized over time, leading to increased tolerance of substandard care [58] and neglect [14], which normalizes [56] and legitimizes neglect [16,62], and, in a worst-case scenario, the acceptance of practices that have been characterised as “evil” [41] or “shitty nursing” [62]. In line with this, Phelan suggests that the neglect of residents in nursing homes should be a question of human rights to stimulate staff reaction to poor practices [1]. 

### 4.5. Study Limitations

Recruitment for our study was influenced by the COVID-19 lockdown, as there were times when nursing home staff were far less accessible than they would have been under normal circumstances. Therefore, we were unable to work systematically enough with theoretical sampling to be able to confidently claim that we have reached saturation. Despite this theoretical shortcoming, we managed to recruit participants from a variety of nursing homes in addition to participants who had quit their jobs in such institutions. Through our application of an iterative process of moving back and forth between gathering and analysing data, we were still able to identify processes that are crucial to continuing neglect across the sampled institutions. 

## 5. Conclusions

In principle, the participants in this study were positive about reporting neglectful practices, based on the assumption that reporting is a way to improve practice. Despite this, there are several obstacles and circumstances that interfere with the expected reporting of neglect in Norwegian nursing homes. Our core finding is that there is a reporting practice and culture that renders neglect invisible in nursing homes. This prevails when nursing staff omit to report neglect in the existing reporting system, and when nursing staff apply staff-protecting strategies. These strategies result in challenges in both the recognition and acknowledgement of neglect, as well as subsequent communication about and reporting of neglect. These reporting practices make neglect invisible in nursing homes and ensure that the invisibility of neglectful practices is maintained, thus compromising the safety and quality of care for residents in nursing homes for the foreseeable future. 

## Figures and Tables

**Figure 1 healthcare-11-01415-f001:**
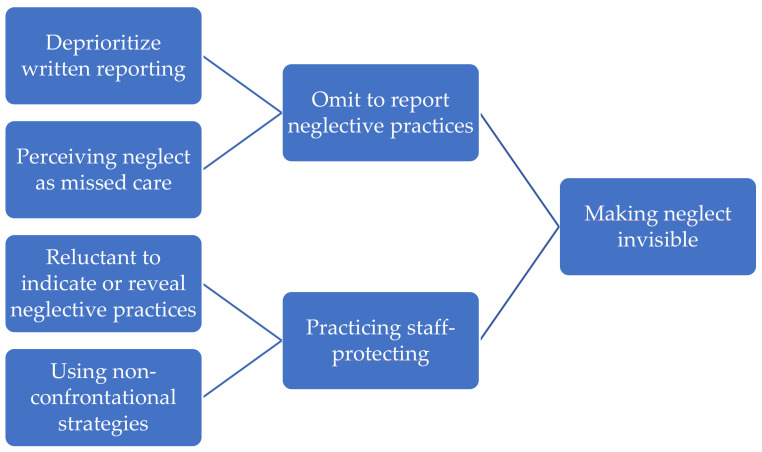
Diagram illustrating the relationships between subcategories, categories, and core finding.

**Table 1 healthcare-11-01415-t001:** An example from the coding process.

Initial Codes	Focused Codes	Subcategory	Category
Lacking knowledge about reporting in IRS Complicated reporting system Lacking time for written reporting Avoiding reporting in IRS if no consequences for the resident Unsure what qualifies as neglect Discrepancies in what is acknowledged as neglect Easier to report to colleague directly Possible to improve if given direct feedback Making colleague accountable through direct feedback Emphasizing communication between staff Wanting to contain errors among staff Receiving no reaction to written reports Receiving no/negative feedback from manager on reports in IRS Preferring to report verbally to ward-nurse/manager Reporting in patient records Different perspectives on adequate care Perceiving neglect as forgotten care Ranking neglect as insignificant/unimportant Perceiving neglect as minor omissions Accepting neglect if it only occurs rarely Reporting as missed nursing care on to-do list Reporting as missed nursing care in patient records Neglecting to report care activities not done	Lacking resources and skills Experiencing obstacles Challenging to report Feelings of ambiguity (in what/how to report) Preferring and valuing direct verbal reporting Receiving insufficient reaction/feedback Choosing to report via informal systems Perceiving neglect as omissions Depending on severity and frequency Not acknowledging neglect Trivializing neglect reporting	Deprioritize written reporting Perceiving neglect as missed care.	Omit to report neglective practices

## Data Availability

The data in this study is available on request from the corresponding author.

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
