# Peer review of "Making Neglect Invisible: A Qualitative Study among Nursing Home Staff in Norway"

_healthcare, 2023, doi:10.3390/healthcare11101415_

Round 1

Reviewer 1 Report

The qualitative study titled "Making neglect invisible: A qualitative study among nursing home staff in Norway" explores the experiences of nursing home staff in Norway regarding the identification and reporting of neglect. Neglect is a serious issue in nursing homes, and this study sheds light on the challenges faced by staff in identifying and reporting cases of neglect, as well as the factors that contribute to neglect remaining invisible.

The study is based on in-depth interviews with nursing home staff and is informed by a grounded theory approach. The authors use this approach to generate a theory about how neglect becomes invisible in nursing homes and the various factors that contribute to this phenomenon. The study is important because it contributes to our understanding of how neglect is perpetuated in nursing homes and the factors that need to be addressed to improve the quality of care for residents.

The study is well-written and the authors have used a rigorous methodology to generate their findings. However, some minor points could be improved. Overall, this study is an important contribution to the literature on neglect in nursing homes and has the potential to inform policy and practice in this area.

L64: the sentence seems incomplete, rewrite, please.

 L88: there is a type, adjust-lease, 

L112: “§196 of the Penal Code “, adjust, please.

Reviewer 2 Report

Thank you for permitting me to review your manuscript, which once the suggested corrections are accepted and made, will make a valuable contribution to research in your field.

Please take time to proofread your manuscript thoroughly there are inconsistencies in tenses, grammar, punctuation, and language throughout the document. I have heavily commented on the introduction. However, the remainder of the paper requires equal scrutiny.

You make no comment or provide no evidence of having gained or receiving ethical approval for the research. Without such evidence, this manuscript cannot be published.

I also find it interesting that you make no reference to nursing professional registration and the requirements 'to do no harm' in your manuscript.

Finally, no commentary has been provided regarding the role of graduate education and continuing education can offer regarding the delivery of safe quality ethical care. A fundamental requirement of any person receiving health or social care. Consideration of this factor may be worthwhile.

Reviewer 3 Report

The authors conducted an  investigation into a significant aspect of nursing care in order to improve the quality of care for residents in nursing homes. Despite the fact that the contents are relevant, I have serious concerns about the article's structure, methodological quality and reporting. Further, editing for English language, style and soundness is required. Please, find attached my specific comments. 

Reviewer 4 Report

It was a pleasure to read this manuscript. This qualitative study addresses serious and sensitive issues common to many developed countries struggling with an ever increasing number of elderly people in need of residential care which is inadequately funded. The result of this is that staff are over-worked regularly, leading to the issues addressed in this research.  The paper describes clearly the qualitative study research question, methods, results and conclusions.  It is written in high-quality scientific English and presents clear implications and steps the staff should take to address the issues raised.  However, it acknowledges the time and financial constraints that need to be addressed if the staff are to action the advice in the conclusions.

1. Perhaps a point to really emphasise in the discussion is the need for the staff to have more time to achieve all the care tasks fully, and this means more funding.  Can this be woven into the discussion and summarised in the conclusion? 

2. Congratulations on the recent publication based on the same qualitative data.  It might be worth making a clear statement in the introduction on how this current analysis differs, to avoid any confusion. Lund, S. B., Skolbekken, J. A., Mosqueda, L., & Malmedal, W. K. (2023). Legitimizing neglect-a qualitative study among nursing home staff in Norway. BMC Health Services Research23(1), 212.  

3. There is s minor typing mistake in the table 1. Third line of the initial codes has the word Lacking repeated.

Round 2

Reviewer 2 Report

Thank you, the changes you have made strengthen your manuscript, and provide a useful resource for other to consider when researching similar issues. Best wishes

Reviewer 3 Report

Dear Editor,

Thank you very much again for the opportunity to review this manuscript. Although some revisions were made, some of the methodological recommendations that give robustness to the research were not accepted. In particular, the methodological choices were not supported by bibliographic references. Please, see below my punctual responses:

Abstract:

The required adjustments were implemented by the authors.

Introduction:

The required adjustments were implemented by the authors.

Materials and Methods:

- The COREQ checklist needs to be referenced.

- The COREQ checklist needs to be attached to the manuscript. 

- The authors justified their decision to use focus group as a survey         method.  However, this choice was not supported by any bibliographic citation.

- My comment about the interview guide questions was not addressed and the authors' choice was not supported with methodological references from the literature. 

- My comment about the table with participants' characteristics was not addressed and the authors choice was not justified with methodological references from the literature.

- My comment about the discussion section was not addressed. 

- My comment about the study's limitations was not addressed completely. There are some intrinsecal limitations to the research methodology that must be stated.
